# IMPROVED GRADIENT ESTIMATORS FOR STOCHASTIC DISCRETE VARIABLES

## ABSTRACT

In many applications we seek to optimize an expectation with respect to a distribution over discrete variables. Estimating gradients of such objectives with respect to the distribution parameters is a challenging problem. We analyze existing solutions including finite-difference (FD) estimators and continuous relaxation (CR) estimators in terms of bias and variance. We show that the commonly used Gumbel-Softmax estimator is biased and propose a simple method to reduce it. We also derive a simpler piece-wise linear continuous relaxation that also possesses reduced bias. We demonstrate empirically that reduced bias leads to a better performance in variational inference and on binary optimization tasks.

## 1 INTRODUCTION

Discrete stochastic variables arise naturally in many applications including topic modeling, semi-supervised learning, clustering, variational memory addressing and reinforcement learning. In many cases, the objective is to minimize an expectation of a function of discrete random variables with respect to the distribution parameters:

$$\min_{\boldsymbol{\phi}} \mathcal{L}[\boldsymbol{\phi}], \quad \text{where } \mathcal{L}[\boldsymbol{\phi}] = \mathbb{E}_{q_{\boldsymbol{\phi}}}\big[f(\boldsymbol{z})\big] = \sum_{\boldsymbol{z}} q_{\boldsymbol{\phi}}(\boldsymbol{z}) f(\boldsymbol{z}). \tag{1}$$

Here, $\boldsymbol{z}$ is a vector of discrete variables under a $\boldsymbol{\phi}$-parameterized distribution $q_{\boldsymbol{\phi}}(\boldsymbol{z})$. For example, in variational inference, $f(\boldsymbol{z})$ is the variational lower bound and $q_{\boldsymbol{\phi}}(\boldsymbol{z})$ is the approximating posterior.

Eq. (1) is commonly minimized by gradient-based methods, which require estimating the gradient $\partial_{\boldsymbol{\phi}} \mathcal{L}[\boldsymbol{\phi}]$ (see Schulman et al. (2015) for an overview). For continuous random variables whose sampling can be reparameterized as a function of other parameter-independent random variables, the *reparameterization trick* (Kingma & Welling (2013), Rezende et al. (2014)) gives low variance gradient estimates. However, this trick is not directly applicable to discrete variables due to the discontinuous cumulative distribution function (CDF).

Extensions of the reparameterization trick to discrete distributions can be grouped into finite-difference (FD) estimators, and continuous relaxation (CR) estimators. We present a summary of three recent FD estimators (Tokui & Sato (2017); Yin & Zhou (2018); Lorberbom et al. (2018) to guide practitioners to the salient aspects of their bias-variance tradeoff and computational complexity. Most importantly, we propose a scalable (but still unbiased) variant of Tokui & Sato (2017) that trades decreased computation for slightly increased variance.

We examine CR estimators including the popular Gumbel-Softmax (Jang et al. (2016); Maddison et al. (2016)) in terms of the gradient bias and propose a new method to reduce the bias of all CR estimators for both binary and categorical variables. Based upon this understanding, we develop a piecewise-linear estimator for binary and categorical variables that is simpler and less biased than Gumbel-Softmax.

Lastly, we provide empirical evidence that these improved estimators allow for faster optimization when training variational autoencoders and when optimizing a continuous relaxation of a combinatorial optimization problem.

### 1.1 RELATED WORK

The most generic approximator of $\partial_\phi \mathcal{L}[\phi]$ is the score function (SF) estimator (a.k.a. REINFORCE Williams (1992), Glynn (1990)). SF suffers from high variance and many remedies have been proposed to reduce this variance (Mnih & Gregor (2014); Gregor et al. (2013); Gu et al. (2015); Mnih & Rezende (2016); Tucker et al. (2017); Grathwohl et al. (2017)). Unbiased estimators that require multiple function evaluations have also been proposed Tokui & Sato (2017), Titsias & Lázaro-Gredilla (2015), Lorberbom et al. (2018), Yin & Zhou (2018). These estimators have lower variance but can be computationally demanding. CR estimators often trade bias for variance and previous CR proposals include straight-through estimators (Bengio et al. (2013); Raiko et al. (2014)), the Gumbel-Softmax estimator (Jang et al. (2016); Maddison et al. (2016)), and overlapping smoothing (Vahdat et al. (2018b;a); Rolfe (2016)).

## 2 THE REPARAMETERIZATION AND MARGINALIZATION ESTIMATOR

We begin a summary of FD estimators with the reparameterization and marginalization (RAM) method of Tokui & Sato (2017). For a single binary variable the expectation in Eq. (1) is enumerated as $\mathcal{L} = \sum_z q_\phi(z) f(z) = q_\phi f(1) + (1 - q_\phi) f(0)$, where $q_\phi \equiv q_\phi(z = 1)$. The gradient is:

$$\partial_\phi \mathcal{L} = \partial_\phi q_\phi (f(1) - f(0)) = \partial_\phi l_\phi \, q_\phi (1 - q_\phi)(f(1) - f(0)), \tag{2}$$

where $q_\phi = \sigma(l_\phi) = (1 + e^{-l_\phi})^{-1}$ and $l_\phi = \text{logit}(q_\phi)$. This derivative involves two function evaluations and contains a finite-difference of $f(z)$. Eq. (2) is an unbiased zero-variance estimate since the summation over $z$ is done explicitly. The generalization to $M$ factorially-distributed random variables $q_\phi(\boldsymbol{z}) = \prod_{i=1}^M q_{\phi,i}(z_i)$ is

$$\partial_\phi \mathcal{L} = \sum_i \partial_\phi q_{\phi,i} \sum_{\boldsymbol{z}_{\backslash i}} q_\phi(\boldsymbol{z}_{\backslash i})(f(z_i = 1, \boldsymbol{z}_{\backslash i}) - f(z_i = 0, \boldsymbol{z}_{\backslash i})), \tag{3}$$

where again the summation over $z_i$ is performed exactly and $q_{\phi,i} \equiv q_{\phi,i}(z_i = 1)$. The summation over $\boldsymbol{z}_{\backslash i}$ can be estimated with a single sample but the derivative requires $M+1$ function evaluations. This limits the applicability of RAM to moderately-sized models. Note that both $f(z_i = 0, \boldsymbol{z}_{\backslash i})$ and $f(z_i = 1, \boldsymbol{z}_{\backslash i})$ are evaluated at the same $\boldsymbol{z}_{\backslash i}$ which leads to a lower-variance estimator. For hierarchical $q_\phi(\boldsymbol{z}) = \prod_{i=1}^M q_{\phi,i}(z_i | \boldsymbol{z}_{<i})$ the derivative takes the form:

$$\partial_\phi \mathcal{L} = \sum_i \sum_{\boldsymbol{z}_{<i}} q_\phi(\boldsymbol{z}_{<i}) \partial_\phi q_{\phi,i} \left[ \sum_{\boldsymbol{z}_{>i}} q_\phi(\boldsymbol{z}_{>i} | 1, \boldsymbol{z}_{<i}) f(\boldsymbol{z}_{>i}, 1, \boldsymbol{z}_{<i}) - \sum_{\boldsymbol{z}_{>i}} q_\phi(\boldsymbol{z}_{>i} | 0, \boldsymbol{z}_{<i}) f(\boldsymbol{z}_{>i}, 0, \boldsymbol{z}_{<i}) \right]. \tag{4}$$

where $q_{\phi,i} \equiv q_{\phi,i}(1 | \boldsymbol{z}_{<i})$. The key insight of Tokui & Sato (2017) is to use common random variates $\boldsymbol{z}_{>i}$ when sampling from $q_\phi(\boldsymbol{z}_{>i} | 1, \boldsymbol{z}_{<i})$ and $q_\phi(\boldsymbol{z}_{>i} | 0, \boldsymbol{z}_{<i})$. This reduces both the variance and the computational cost. Tokui & Sato (2017) show that this estimator is optimal because it exactly sums over the binary variables whose probability distribution is being differentiated.

A RAM estimator can also be constructed for categorical variables. For a single one-hot encoded categorical variable $y = (y^0, ...y^{A-1}), y^a \in \{0, 1\}, \sum_a y^a = 1$, the derivative of Eq. (1) is

$$\partial_\phi \mathcal{L} = \partial_\phi \sum_y q_\phi(y) f(y) = \sum_a (\partial_\phi l_\phi^a) \sum_b q_\phi^a q_\phi^b (f^a - f^b), \tag{5}$$

where $f^a = f(y^a = 1)$, and $q_\phi^a = e^{l_\phi^a} / \sum_b e^{l_\phi^b}$. The generalization to many categorical variables proceeds as for binary variables. For example, the derivative of a factorial categorical distribution over $\boldsymbol{y} = \{y_i^a \mid 0 \le a < A, 1 \le i \le M\}$ is

$$\partial_\phi \mathcal{L} = \sum_i \sum_{\boldsymbol{y}_{\backslash i}} q_\phi(\boldsymbol{y}_{\backslash i}) \sum_{a,b} (\partial_\phi l_{\phi,i}^a) q_{\phi,i}^a q_{\phi,i}^b \big( f(y_i^a = 1, \boldsymbol{y}_{\backslash i}) - f(y_i^b = 1, \boldsymbol{y}_{\backslash i}) \big). \tag{6}$$

This derivative can again be estimated with a single sample but requires $MA$ function evaluations. Since RAM is unbiased and has the minimal variance (due to the explicit summation over the differentiated variable), we use it as a baseline to evaluate computationally cheaper alternatives.

## 2.1 Sampled Reparameterization and Marginalization

We propose a modification to RAM that allows us to trade decreased computational cost for increased variance. For binary variables, $\partial_\phi q_{\phi,i} = q_{\phi,i}(1 - q_{\phi,i})\partial_\phi l_{\phi,i}$ where $q_{\phi,i} = \sigma(l_{\phi,i})$, so that each term in Eq. (3) or (4) is proportional to $q_{\phi,i}(1 - q_{\phi,i})$. In many applications, we observe that the distribution of a large number of variables ($q_{\phi,i}$) are drawn to 0 or 1 early during optimization. Such variables have negligible contribution to the full derivative. We exploit this observation to reduce the computational cost by including variable $z_i$ in the full gradient with probability $p_i = 4q_{\phi,i}(1 - q_{\phi,i})/\beta$, where $\beta$ is an adjustable hyperparameter. This means that we replace the derivative in Eq. (3) with

$$\partial_\phi \mathcal{L} = E_{\xi \sim p}\Big[\sum_i (\partial_\phi l_{\phi,i})\frac{\beta \xi_i}{4}\sum_{\bm{z}_{\setminus i}} q_\phi(\bm{z}_{\setminus i})\big(f(z_i = 1, \bm{z}_{\setminus i}) - f(z_i = 0, \bm{z}_{\setminus i})\big)\Big], \qquad (7)$$

where $\xi_i \in \{0, 1\}$ are Bernoulli variables with probabilities $p_i$ indicating whether $z_i$ is included or not. We evaluate Eq. (7) by sampling $\xi$ and only then evaluating non-zero terms. In Section 4 we show that in the context of variational inference on MNIST the number of function evaluations is reduced by an order of magnitude while still allowing for effective optimization. As always, this computational saving comes at the cost of increased gradient variance which slows training.

In the categorical case, each term in Eq. (6) is accompanied by $q_{\phi,i}^a q_{\phi,i}^b$ that can be used to assign importance to the $(a, b)$ edge of the simplex for variable $y_i$. The computational cost $MA$ can be reduced by keeping each term with probability $p_i^{a,b} = 4q_{\phi,i}^a q_{\phi,i}^b/\beta$.

Appendices A and B summarize two other FD estimators, ARGMAX Lorberbom et al. (2018) and ARM Yin & Zhou (2018), by noting their bias-variance tradeoff and computational complexity.

## 3 Continuous Relaxation Estimators

Unlike FD estimators that use multiple function evaluations to approximate the gradient, continuous relaxation estimators extend the reparameterization trick to discrete variables by approximating them with continuous variables: $z \to \zeta = \zeta_\beta(\rho, q_\phi)$, where $\beta$ is a parameter that controls the approximation, $\rho \in \mathcal{U}[0, 1]$ is uniform random variable and $q_\phi$ are parameterized probabilities.[1] We use the same symbol $\zeta$ to denote the random variable and its reparameterization by $\rho$. The objective function $\mathcal{L}[\phi] = \sum_{\bm{z}} q_\phi(\bm{z})f(\bm{z})$ is replaced with $\tilde{\mathcal{L}}[\phi] = \mathbb{E}_{\bm{\rho}}\big[f\big(\bm{\zeta}(\rho, q_\phi)\big)\big]$ and its gradients are computed using the chain rule

$$\partial_\phi \tilde{\mathcal{L}}[\phi] = \mathbb{E}_{\bm{\rho}}\left[\sum_i \partial_{\zeta_i} f(\bm{\zeta})(\partial_{q_i}\zeta_i)(\partial_\phi q_i)\Big|_{\zeta_i(\rho_i, q_{\phi,i}(1|\bm{\zeta}_{<i}))}\right] \qquad (8)$$

These gradients can be computed efficiently by automatic differentiation libraries. However, because the objective function is changed, CR estimators in Eq. (8) are biased. Nevertheless, in practice the bias is often small enough to allow for effective optimization.

### 3.1 The Gumbel-Softmax Estimator

The most popular CR estimator is Gumbel-Softmax (GSM) (Jang et al. (2016); Maddison et al. (2016))[2]. For binary variables, this relaxation takes the simple form

$$\zeta_i(\rho_i, q_i) = \sigma\left(\beta\big[\sigma^{-1}(q_i) + \sigma^{-1}(\rho_i)\big]\right), \qquad (9)$$

---

[1]Continuous relaxations have been instrumental in constructing advanced control variate for SF estimates in Tucker et al. (2017) and Grathwohl et al. (2017). We discuss the REBAR estimator from Tucker et al. (2017) in Appendix E.

[2]More precisely, Maddison et al. (2016) considered problems in variational inference where $f(\bm{z})$ is relaxed by replacing a generative distribution with its continuous relaxation. In contrast, Jang et al. (2016) directly relaxed discrete variables $z \to \zeta$ without changing the objective, thus replacing $f(\bm{z}) \to f(\bm{\zeta})$. This does not produce a consistent objective for a probabilistic model, but as Jang et al. (2016) have shown, works well in practice. We work with generic $f(\bm{z})$ and thus only consider the approach of Jang et al. (2016).

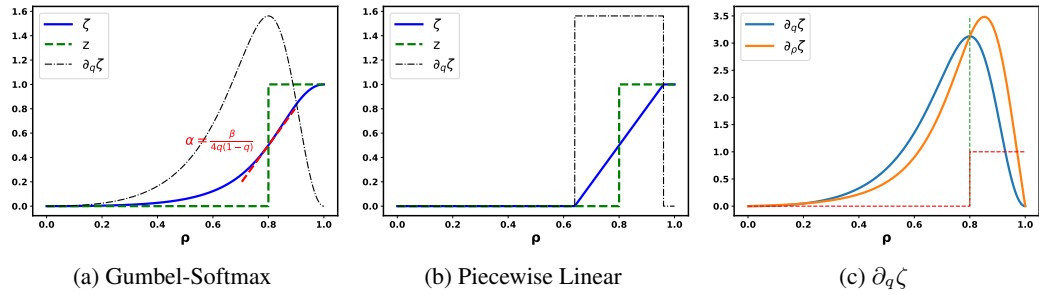

(a) Gumbel-Softmax    (b) Piecewise Linear    (c) $\partial_q \zeta$

Figure 1: Continuous relaxations at $\beta = 2$ to a Bernoulli random variable $\text{Ber}(q_\phi = 0.8)$: (a) shows the GSM approximation and (b) shows a piece-wise linear approximation to the discontinuous CDF. (c) shows the effect of a modification to GSM that reduces its bias.

where $q_i \equiv q_{\phi,i}(1|\boldsymbol{\zeta}_{<i})$. This relaxation and its derivative $\partial_q \zeta$ are shown in Fig. 1(a). $\beta$ controls the sharpness of the relaxation and tunes the trade-off between the bias (closeness of $\zeta$ to $z$) and variance of $\partial_q \zeta$. We note that the slope of the relaxed $\zeta(\rho)$ at $\zeta = 1/2$ is $\alpha \equiv \beta/[4q(1-q)]$ and thus becomes large when $q$ approaches 0 or 1.

## 3.2 Improved Continuous Relaxation Estimators

In this section, we analyze the bias introduced by Eq. (8) and propose a simple method to reduce it. The bias of $\mathbb{E}_{\boldsymbol{\rho}}\left[\partial_{\zeta_i} f(\boldsymbol{\zeta})\partial_\phi \zeta_i\right]$ has two sources: (a) the relaxation of $\zeta_j$ for $j \neq i$ and (b) the relaxation of $\zeta_i$. To characterize the latter bias, we start with a single binary variable and write the gradient as the following integral:

$$\partial_\phi \mathcal{L} = \partial_\phi q_\phi \big(f(1) - f(0)\big) = \partial_\phi q_\phi \int_0^1 d\xi\, \partial_\xi f(\xi) = \partial_\phi q_\phi \int_0^1 d\rho\, \frac{\partial \xi}{\partial \rho}\partial_\xi f(\xi). \tag{10}$$

Here, $\xi(\rho)$ is *any* continuous function satisfying $\xi(0) = 0$ and $\xi(1) = 1$. For a non-decreasing function $\xi(\rho)$, we can view $\xi$ as a random variable with inverse CDF $\xi(\rho)$ and $\rho$ as a uniform random variable $\rho \in \mathcal{U}[0,1]$. With this interpretation the gradient can be written as an expectation

$$\partial_\phi \mathcal{L} = \partial_\phi q_\phi\, \mathbb{E}_\rho\left[\frac{\partial \xi}{\partial \rho}\partial_\xi f(\xi)\right], \tag{11}$$

which can be estimated by sampling. If $\partial_\xi f(\xi)$ does not vary significantly in the interval $\xi \in [0,1]$ then the variance of this estimate is controlled by $\text{var}(\partial \xi/\partial \rho) = \int_0^1 d\rho\, (\partial \xi/\partial \rho)^2 - 1$. Thus, the more non-linear $\xi(\rho)$ is, the higher will be the variance of estimate Eq. (11). This idea can be extended to factorial $q_\phi(\boldsymbol{z}) = \prod_{i=1}^M q_{\phi,i}(z_i)$ as in Eq. (3):

$$\partial_\phi \mathcal{L} = \sum_i \partial_\phi q_{\phi,i} \sum_{\boldsymbol{z}_{\backslash i}} q_\phi(\boldsymbol{z}_{\backslash i}) \int d\rho_i \frac{\partial \xi}{\partial \rho_i}\partial_{\xi_i} f(\xi, \boldsymbol{z}_{\backslash i}). \tag{12}$$

At this point there is no relationship between $\xi_i$ and $z_i$, and Eq. (12) is just a higher variance version of Eq. (3). However, if we relax $\boldsymbol{z}_{\backslash i} \to \boldsymbol{\zeta}_{\backslash i}(\rho, q_\phi)$ and choose $\xi_i(\rho_i) = \zeta_i(\rho_i, q_{\phi,i})$ we obtain a *biased* estimator

$$\partial_\phi \mathcal{L} \approx \sum_i \mathbb{E}_{\boldsymbol{\rho}}\left[\partial_{\zeta_i} f(\boldsymbol{\zeta})\frac{\partial \zeta_i}{\partial \rho_i}\partial_\phi q_{\phi,i}\right], \tag{13}$$

where the bias comes from the deviation of $\boldsymbol{\zeta}_{\backslash i}$ from $\boldsymbol{z}_{\backslash i}$ in the function evaluations. In other words, Eq. (13) uses an unbiased form for the differentiated variable $\zeta_i$ and the only bias comes from relaxing the remaining variables $\boldsymbol{\zeta}_{\backslash i}$.

The variance of each term is controlled by $\text{var}\left(\frac{\partial \zeta_i}{\partial \rho_i}\right)$ and is reduced by making $\zeta_i(\rho_i, q_{\phi,i})$ more linear. The bias is reduced by making $\boldsymbol{\zeta}$ closer $\boldsymbol{z}$. Varying $\zeta_i(\rho_i, q_{\phi,i})$ between linear (low variance) and step function (low bias) allows for a controllable trade-off. We call Eq. (13) the improved continuous relaxation (ICR) estimator. The original CR estimator of Eq. (8) for factorial $q_\phi$ has the form

$$\partial_\phi \tilde{\mathcal{L}}[\phi] = \sum_i \mathbb{E}_{\boldsymbol{\rho}}\left[\partial_{\zeta_i} f(\boldsymbol{\zeta})\frac{\partial \zeta_i}{\partial q_i}\partial_\phi q_{\phi,i}\right], \tag{14}$$

and comparing this to the ICR estimate in Eq. (13), we see that CR can be transformed into ICR by the replacing $\partial_{q_i} \zeta_i$ with $\partial_{\rho_i} \zeta_i$. This change can be simply implemented using TensorFlow's stop_gradient $\equiv$ **sg** notation by replacing

$$\zeta_i(\rho_i, q_i) \quad \text{with} \quad \zeta_i\big(\rho_i + q_i - \mathbf{sg}(q_i), \mathbf{sg}(q_i)\big) \tag{15}$$

in Eq. (8). We emphasize that the ICR estimator in Eq. (13) is less biased than the direct CR estimator because, in the case of a single variable, ICR is unbiased while CR is not. Further, this decrease in bias is not accompanied by an increase in variance. We show in Appendix C that similar benefits are obtained for hierarchical $q_\phi$. Finally, we note a conceptual relationship between sampled RAM and ICR: both estimators evaluate the gradient only through a subset of variables $z_i$. Sampled RAM chooses this subset explicitly and evaluates the gradients with FD while ICR samples relaxed variables where only a subset of them will possess non-negligible gradients.

## 3.3 PIECE-WISE LINEAR RELAXATION

Inspired by ICR and a better understanding of bias-variance trade-off, we propose a piece-wise linear relaxation (PWL) depicted in Fig. 1(b). The linear part is centered at $\rho = 1 - q$ so that the corresponding binary variable is obtained by $z = \text{round}(\zeta)$. The slope is given by $\alpha = \beta/[4q(1-q)]$ similar to the Gumbel-Softmax slope [3]. The explicit expression for PWL smoothing is

$$\zeta(\rho, q) = \big[0.5 + \alpha(\rho - (1 - q))\big]_0^1, \tag{16}$$

where $[x]_0^1 \equiv \min(1, \max[0, x])$ is the hard sigmoid function (note that $\partial_q \alpha = 0$). This relaxation has several attractive properties. Firstly, we have $\partial_{q_i} \zeta_i = \partial_{\rho_i} \zeta_i$, which means that the CR and ICR estimators coincide for PWL. Secondly, PWL has easily interpretable expressions for bias and variance. In the case of a function of a single variable the variance is given by $\text{var}(\partial_\rho \zeta) = \alpha - 1$ while the bias in computing expectation $\sum_z f(z)$ over the relaxed distribution is $\int d\rho f(\zeta(\rho)) - \sum_z f(z) = \{\int_0^1 dx f(x) - [f(0) + f(1)]/2\}/\alpha$. This clearly shows that $\alpha$ trades bias for variance.

Finally, the PWL relaxation defined in Eq. (16) can be considered as the inverse CDF of the PDF

$$q(\zeta) = (1 - \epsilon)\left[(1 - q)\delta(\zeta) + q\delta(\zeta - 1)\right] + \epsilon\,\mathcal{U}[0, 1], \quad \text{where} \quad \epsilon = [4q(1 - q)]/\beta. \tag{17}$$

Eq. (17) is a mixture of two delta distributions centered at zero and one, and a uniform distribution defined in the interval $[0, 1]$. Samples from $q(\zeta)$ are in the continuous interval with probability $[4q(1 - q)]/\beta$, and $\zeta = 0/1$ have probability proportional to the probability of the binary states.

## 3.4 CATEGORICAL PIECE-WISE LINEAR RELAXATION

We now extend ICR estimators to categorical variables. For a single categorical variable we apply the integral representation to each edge $(a, b)$ of the simplex in Eq. (5) and relax this pair of variables using PWL: $y \to y^{a,b} = \{y^a = \big[0.5 + \alpha^{a,b}\big(\rho^{a,b} - q^b/(q^a + q^b)\big)\big]_0^1, y^b = 1 - y^a, y^{c \neq a,b} = 0\}$ where $\rho^{a,b} \sim \mathcal{U}[0, 1]$ and $\alpha^{a,b}$ is the slope. We replace the summation over the edges of the simplex by sampling one edge at a time with probability $p^{a,b} = (q^a + q^b)/(A - 1)$. Details are found in Appendix D and we provide the final result:

$$\mathcal{L} = \mathbb{E}_{(a,b) \sim p^{a,b}}\left[\mathbb{E}_{\rho \in \mathcal{U}[0,1]}\left[f(\tilde{y}^{a,b})\right]\right], \tag{18}$$

where $\tilde{y}^{a,b}$ has the same value as $y^{a,b}$ but has the gradient scaled by $\gamma^{a,b} = (A - 1)(q^a + q^b)$.[4] The probabilities of edge selection and the scale factor are chosen to give correct values for the objective and its gradient. Extension of this categorical PWL estimator to multivariate distributions is straightforward; For example, for factorial distributions one relaxes each categorical variable $y_i$ by sampling an edge with probability $p_i^{a,b} = (q_i^a + q_i^b)(A - 1)$ and uses a single function evaluation. The resulting gradient is unbiased for a single variable and introduces a bias in the multivariate case which makes it an ICR estimator.

---

[3]More generally, the slope of the linear part $\alpha$ can be chosen arbitrarily as long as $\alpha \geq 0.5/\min(q, 1 - q)$ so that both $\zeta = 0$ and $\zeta = 1$ have non-zero probability.

[4]In Tensorflow notation $\tilde{y}^{a,b} \equiv \mathbf{sg}(y^{a,b}) + \mathbf{sg}(\gamma^{a,b})(y^{a,b} - \mathbf{sg}(y^{a,b}))$.

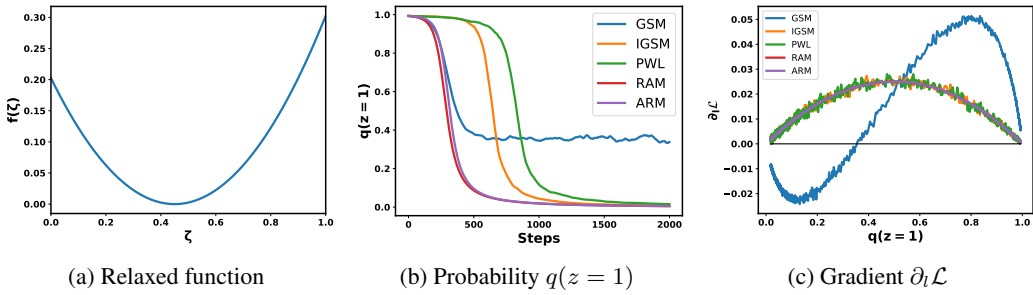

(a) Relaxed function    (b) Probability $q(z=1)$    (c) Gradient $\partial_l \mathcal{L}$

Figure 2: Convex, single binary-variable toy example: (a) The relaxed objective $f(\zeta) = (\zeta - 0.45)^2$. (b) The probability $q_\phi(z=1)$ during optimization. (c) The gradient $\partial_l \mathcal{L}$ of all estimators; the bias of GSM prevents proper minimization.

### 3.5 IMPROVED CATEGORICAL GUMBEL-SOFTMAX ESTIMATOR

The sampling done in the categorical PWL estimator leads to increased variance of the gradients. As an alternative, we suggest an improved version of the categorical Gumbel-Softmax estimator. Recall that the Gumbel-Softmax estimator for the categorical case has the form:

$$\zeta^a(\rho; \boldsymbol{q}) = \text{softmax}(\beta (\log q^a + \log \rho^a)), \quad \text{where} \quad \rho^a = \frac{\log u^a}{\sum_b \log u^b}, \; u^b \sim \mathcal{U}[0,1]. \quad (19)$$

In Eq. (20), we propose a new version of this estimator by applying the same trick as in Eq. (15). This estimator remains biased but we empirically demonstrate that its bias is reduced.

$$\zeta(\rho, \boldsymbol{q}) \to \zeta(\rho + \boldsymbol{q} - \mathbf{sg}(q), \mathbf{sg}(q)). \quad (20)$$

## 4 EXPERIMENTS

In this section we compare the FD and CR estimators and their improved variants on a number of examples. We start with one-variable toy examples that illustrate the bias of GSM estimator and then move to the training of variational auto-encoders and a combinatorial optimization problem.

### 4.1 TOY EXAMPLE

We begin with an illustrative single-variable example (Tucker et al. (2017)) with objective $\mathcal{L} = \sum_z q_\phi(z) f(z)$ where $f(z) = (z - 0.45)^2$. The relaxed convex function $f(\zeta)$ depicted in Fig. 2(a) has two local maxima, one of which is the minimum over the discrete domain. We compare five gradient methods: RAM, Eq. (2); ARM (see Appendix B); PWL, Eq. (16); GSM, Eq. (9); and improved Gumbel-Softmax (IGSM), Eq. (15) (see Appendix F for experimental details). Fig. 2(b) shows the evolution of $q_\phi(z=1)$ during training which demonstrates the bias associated with GSM. To quantify this bias, we plot the value of the gradient of all estimators for different values of $q_\phi(z=1)$ in Fig. 2(c). We observe that the GSM gradient has the wrong sign for a large interval in $q_\phi(z=1)$ which prevents GSM from converging to the true minimum.

To understand the nature of the bias that GSM introduces, we plot the derivatives $\partial_q \zeta$ and $\partial_\rho \zeta$ corresponding to GSM and IGSM respectively in Fig. 1(c). We see that $\partial_q \zeta$ is biased towards the value of $z = \text{round}(\zeta)$ that has the highest probability. This means that GSM in Eq. (14) will oversample the derivative $\partial_\zeta f(\zeta)$ from the most probable mode. In example Fig. 2(a) this results in oversampling the derivative from $z = 0$ mode which creates a gradient that pushes optimization away from the true minimum $z = 0$. The bias is reduced as $\beta$ is increased.

In Appendix I.1 we consider an example for a concave function over a binary variable and observe similar effects. We also consider both concave and convex functions over a categorical variable in Appendix I.2. In all scenarios, the bias of GSM prevents its convergence to the correct minimum.

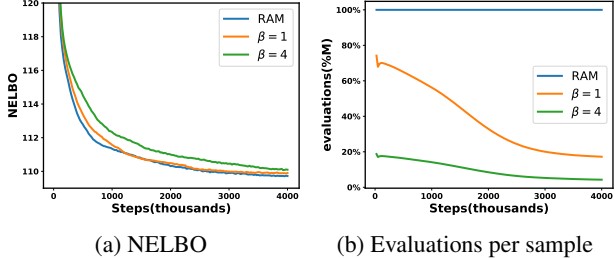

|  (a) NELBO | (b) Evaluations per sample |

Figure 3: NELBO of the RAM and sampled RAM estimators on MNIST trained using the $200H - 784V$ architecture having a linear decoder: (a) plots the decrease in training NELBO for RAM and two variants of sampled RAM. (b) shows the computational savings of sampled RAM.

## 4.2 DISCRETE VARIATIONAL AUTOENCODERS

Next, we test the estimators by training variational autoencoders Kingma & Welling (2013) with discrete priors. The objective is the negative expectation lower bound on the log-likelihood (NELBO):

$$\mathcal{L}[\boldsymbol{\phi}, \boldsymbol{\theta}] = \sum_{\boldsymbol{x} \sim \text{data}} \mathbb{E}_{\boldsymbol{z} \sim q_{\boldsymbol{\phi}}(\boldsymbol{z}|\boldsymbol{x})} \left[ f_{\boldsymbol{\theta}, \boldsymbol{\phi}}(\boldsymbol{z}, \boldsymbol{x}) \right], \quad \text{where} \quad f_{\boldsymbol{\theta}, \boldsymbol{\phi}}(\boldsymbol{z}, \boldsymbol{x}) = -\log \frac{p_{\boldsymbol{\theta}}(\boldsymbol{z}) p_{\boldsymbol{\theta}}(\boldsymbol{x}|\boldsymbol{z})}{q_{\boldsymbol{\phi}}(\boldsymbol{z}|\boldsymbol{x})}. \quad (21)$$

Here, $p_{\boldsymbol{\theta}}(\boldsymbol{z})$ is the prior, $p_{\boldsymbol{\theta}}(\boldsymbol{x}|\boldsymbol{z})$ is the decoder, and $q_{\boldsymbol{\phi}}(\boldsymbol{z}|\boldsymbol{x})$ is the approximating posterior. $\mathcal{L}$ is minimized with respect to the $\boldsymbol{\theta}$ parameters of the generative model and the $\boldsymbol{\phi}$ parameters of the approximate posterior. The latter minimization corresponds to Eq. (1) and we can apply the various estimators to propagate $\boldsymbol{\phi}$-derivatives through discrete samples $\boldsymbol{z}$. For CR estimators we replace $\boldsymbol{z}$ with $\boldsymbol{\zeta}$ to compute $\partial_{\boldsymbol{\phi}} \mathcal{L}[\boldsymbol{\phi}, \boldsymbol{\theta}] \approx \sum_{\boldsymbol{x} \sim \text{data}} \mathbb{E}_{\rho} [\partial_{\boldsymbol{\phi}} f_{\boldsymbol{\theta}, \boldsymbol{\phi}}(\boldsymbol{\zeta}, \boldsymbol{x})]$, where the expectation is evaluated with a single relaxed sample per data point $\boldsymbol{x}$. The $\boldsymbol{\theta}$-derivative can be calculated directly from Eq. (21) using the discrete $\boldsymbol{z}$ variables. Thus, the $\boldsymbol{\phi}$ and $\boldsymbol{\theta}$ derivatives are evaluated separately requiring two passes through the computation graph with either relaxed or discrete samples. Jang et al. (2016) evaluate both derivatives in one pass (using $\boldsymbol{\zeta}$) thereby introducing bias in the $\boldsymbol{\theta}$ derivatives. We refer to these two possibilities as one/two-pass training and compare their performance.

Following Maddison et al. (2016); Jang et al. (2016); Tucker et al. (2017), we consider four architectures with binary variables denoted by $200H - 784V$, $200H - 200H - 784V$, $200H \sim 784V$, and $200H \sim 200H \sim 784V$ (see Appendix F for details). First, we compare the sampled RAM estimator of Eq. (7) on $200H - 784V$ model for different values of $\beta$, where the probability of updating variable $z_i$ is $p_i = 4q_i(1 - q_i)/\beta$. Fig. 3(a) shows the NELBO on the training set. As expected, increased $\beta$ leads to increased gradient variance which slows training. Fig. 3(b) shows the average number of function evaluations performed in Eq. (7). Many units become deterministic early in training leading to significant computational savings with the sampled RAM method.

In Fig. 4(a), we include several CR estimators on the same architecture and observe that the GSM estimator performs significantly worse than other estimators. With linear decoders, the objective function Eq. (21) is convex in $z_i$. Thus, similar to the example in Fig. 2, GSM learns a distribution with higher entropy leading to poorer performance. The entropy of all estimators during training is shown in Fig. 4(b). We note that two-pass training performs better then one-pass training for all estimators. Although two-pass training requires twice the computation in the worst case, this overhead is negligible for these models due to GPU parallelization. We observe that ARM performs poorly confirming that its high variance impedes training. The PWL estimator with two-pass training performs on par with RAM, while being more computationally efficient. Lastly, the REBAR estimator (see Appendix E), using the settings of Tucker et al. (2017), performs on par with PWL thereby indirectly confirming that the relaxed objective is a good control variate for the discrete one. We used $\beta = 2$ in the above experiments, similar to Jang et al. (2016); Maddison et al. (2016). To understand the dependence on $\beta$, we plot final train NELBO in Fig. 4(c). We find that improved estimators are less sensitive to the choice of $\beta$.

In Fig. 5, we repeat the experiments for the non-linear architecture $200H \sim 784V$. The one-pass GSM estimator exhibits instability which is remedied by two-pass training. However, two-pass GSM still performs worse than IGSM and PWL due to its bias. Interestingly, one-pass training works better for IGSM/PWL. We observe this repeatedly in the nonlinear models. Unlike the linear case, the RAM estimator converges faster initially but later in training is outperformed by the higher-variance IGSM, PWL and ARM. It is likely that additional noise prevents the latent units from being

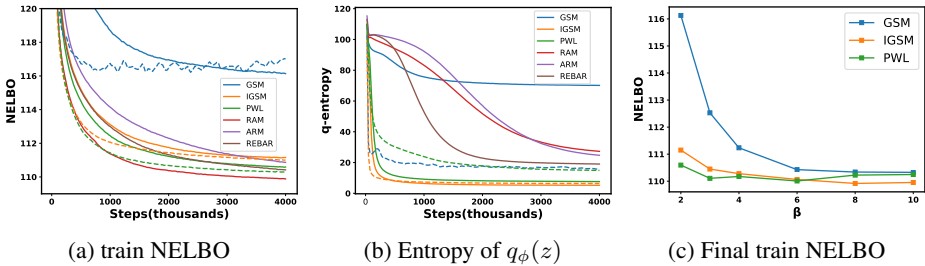

(a) train NELBO          (b) Entropy of $q_\phi(z)$          (c) Final train NELBO

Figure 4: MNIST training on the linear architecture $200H - 784V$: (a) compares training NELBO of CR estimators (all estimators use $\beta = 2$). Solid/dashed lines correspond to one/two-pass training. (b) shows the entropy of the learned posterior approximations; the GSM bias induces much higher entropy. (c) dependence of the final trained NELBO on $\beta$; higher $\beta$ corresponds to lower bias.

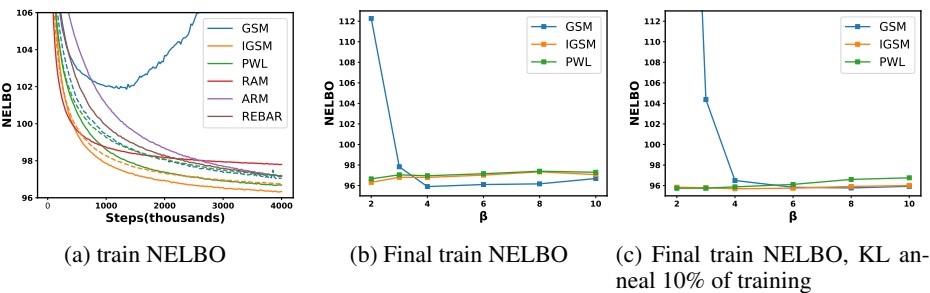

(a) train NELBO          (b) Final train NELBO          (c) Final train NELBO, KL anneal 10% of training

Figure 5: MNIST training on the non-linear architecture $200H \sim 784V$: (a) compares training NELBO of CR estimators (all estimators use $\beta = 2$). Solid/dashed lines correspond to one/two-pass training. (b) Final training NELBO for different $\beta$, (c) Final training NELBO using KL annealing; explicit annealing erases the gains of GSM.

turned off early in training which is known to cause poor performance in VAEs. As in the linear case we plot the dependence of the final train NELBO on $\beta$ in Fig. 5(b). The GSM estimator outperforms IGSM and PWL for $\beta \geq 4$ because its bias favors higher entropy approximating posteriors which inhibit latent units from turning off early in training. A well known resolution for inactive latent units is KL-annealing Bowman et al. (2016). Fig. 5(c) shows that KL annealing indeed improves IGSM and PWL by inhibiting over-pruning of latent units[5]. As with the linear case, the IGSM and PWL estimators are more stable under variations in $\beta$. Here, REBAR underperforms the ICR estimators likely due to its higher variance.

Additional results for other VAE models and for categorical variables on both MNIST and OM-NIGLOT are presented in Appendix I.3 with similar conclusions. The interested reader is referred to Appendix G for experimental results on training encoder part of a VAE with pretrained generative model. Appendix H compares estimators for solving the discrete maximum clique optimization problem analyzed in Patish & Ullman (2018).

## 5 CONCLUSION

We have reviewed several finite-difference (FD) and continuous relaxation (CR) estimators of the gradients of the objective Eq. (1). FD estimators like RAM Tokui & Sato (2017) can give unbiased low variance estimates but often require multiple function evaluations. We proposed a less expensive version of RAM that requires an order of magnitude fewer function evaluations with a controllable decrease in performance. In contrast, CR estimators, like Gumbel-Softmax (GSM), require a single pass through the objective function and can be computed efficiently giving low variance but biased gradients. We analyzed the nature of the bias introduced by CR estimators and proposed a way to reduce it. This gives rise to improved CR (ICR) estimators, like improved GSM and piece-wise linear.

---

[5]In experiments not reported here we observed that the entropy regularizing benefits of the GSM bias can be achieved with explicit entropy regularization of the objective.

These ICR estimators are unbiased for a single variable and less biased for many variables. Experiments on VAE training and discrete optimization confirm the theoretical predictions and illustrate the advantages of lower-bias estimators.

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

# A    ARGMAX

Lorberbom et al. (2018) proposed a FD estimator that we refer to as ARGMAX. For a single variable, ARGMAX relies on an identity that approximates $\partial_\phi \mathcal{L} = \partial_\phi q_\phi[f(1) - f(0)] = (\partial_\phi l_\phi)q_\phi[1 - q_\phi][f(1) - f(0)]$. With $q_\phi = \sigma(l_\phi)$, Lorberbom et al. (2018) show that

$$\sigma(l_\phi)[1 - \sigma(l_\phi)][f(1) - f(0)] = \lim_{\epsilon \to 0} \mathbb{E}_\rho \left[ \frac{\Theta\big(\epsilon f(1) + l_\phi + \sigma^{-1}(\rho)\big) - \Theta\big(\epsilon f(0) + l_\phi + \sigma^{-1}(\rho)\big)}{\epsilon} \right] \tag{22}$$

where $\Theta$ is the Heaviside step function and $\rho \sim \mathcal{U}[0, 1]$.[6] Lorberbom et al. (2018) approximate the right side of Eq. (22) by sampling $\rho$ and evaluating at finite $\epsilon$ which introduces bias and variance. Decreasing $\epsilon$ decreases bias but increases variance.

In the categorical case the gradient of Eq. (5) can correspondingly be written as

$$\partial_\phi \mathcal{L} = \sum_a \partial_\phi l_\phi^a \lim_{\epsilon \to 0} \frac{1}{\epsilon} \mathbb{E}_\rho \left[ \big[ a = \arg\max_b(\epsilon f^b + l_\phi^b + \gamma^b) \big] - \big[ a = \arg\max_b(l_\phi^b + \gamma^b) \big] \right], \tag{23}$$

where $\gamma_b = -\log(-\log(\rho^b))$ are Gumbel variables and $[pred]$ is the indicator function (1 if $pred$ is true and 0 otherwise). This derivative requires $A$ function evaluations, similar to RAM. Lorberbom et al. (2018) extend the single variable result to multivariate distributions similar to Eqs. (3) and (4). ARGMAX has the same computational complexity as RAM, but is biased and has higher variance than RAM. Thus, it is suboptimal to Eqs. (3) and (4) and for this reason we do not perform experiments with ARGMAX.

# B    THE AUGMENT-REINFORCE-MERGE ESTIMATOR

FD estimators are computationally expensive and require multiple function evaluations per gradient. A notable exception is the Augment-REINFORCE-Merge (ARM) estimator introduced in Yin & Zhou (2018). ARM provides an unbiased estimate using only two function evaluations for the factorized multivariate distribution, regardless of the number of variables. For a single binary variable, the ARM derivative is given by

$$\partial_\phi \mathcal{L} = \partial_\phi l \, q_\phi(1 - q_\phi)(f(1) - f(0)) = \partial_\phi l \, \mathbb{E}_{\rho \sim \mathcal{U}[0,1]}[(f(z^{(2)}) - f(z^{(1)}))(\rho - 0.5)],$$
$$z^{(1)} = \Theta(q_\phi - \rho) \quad z^{(2)} = \Theta(\rho - 1 + q_\phi). \tag{24}$$

The expectation is approximated with a single sample $\rho$. This estimator has a significantly lower variance than REINFORCE since the expectation contains the difference $f(z^{(2)}) - f(z^{(1)})$ rather then the function itself. For multivariate $f$ and factorial $q_\phi(\boldsymbol{z})$ the derivative is

$$\partial_\phi \mathcal{L} = \sum_i \partial_\phi l_{\phi,i} \, \mathbb{E}_{\boldsymbol{\rho} \sim \mathcal{U}[0,1]}[(f(z_i^{(2)}, \boldsymbol{z}_{/i}^{(1)}) - f(z_i^{(1)}, \boldsymbol{z}_{/i}^{(1)}))(\rho_i - 0.5)], \tag{25}$$

where $z_i^{(1)} = \Theta(q_{\phi,i} - \rho_i)$ and $z_i^{(2)} = \Theta(\rho_i - 1 + q_{\phi,i})$. Yin & Zhou (2018) observed that one can replace $f(z_i^{(2)}, \boldsymbol{z}_{/i}^{(1)}) \to f(z_i^{(2)}, \boldsymbol{z}_{/i}^{(2)})$ without changing the expectation which allows evaluation by a single sample and two function evaluations $f(\boldsymbol{z}^{(1)}), f(\boldsymbol{z}^{(2)})$ regardless of $M$:

$$\partial_\phi \mathcal{L} = \sum_i \partial_\phi l_{\phi,i} \, \mathbb{E}_{\boldsymbol{\rho} \sim \mathcal{U}[0,1]}[(f(\boldsymbol{z}^{(2)}) - f(\boldsymbol{z}^{(1)}))(\rho_i - 0.5)]. \tag{26}$$

However, this change comes at the cost of higher variance of the expectation[7]. Thus, while ARM estimator provides a low-variance gradient estimate for a single variable, it introduces high variance

---

[6]We motivate this identity by noting that the non-zero contribution comes from the region $\sigma(-\epsilon f(1) - l_\phi) \leq \rho \leq \sigma(-\epsilon f(0) - l_\phi)$ assuming $f(1) > f(0)$. For small $\epsilon$, samples land within this region with probability $p \sim \sigma(l_\phi)(1 - \sigma(l_\phi))\epsilon[f(1) - f(0)]$ giving rise to the identity. The variance of Eq. (22) is $\mathbb{V}\text{ar}\big(\text{Ber}(p)\big)/\epsilon^2 \propto \sigma(l_\phi)(1 - \sigma(l_\phi))[f(1) - f(0)]/\epsilon$ to leading order in $1/\epsilon$.

[7]Denoting $g^{(a)}(\boldsymbol{z}_{/i}) = f(z_i^{(a)}, \boldsymbol{z}_{/i}), a = 1, 2$, we can write

$$\text{VAR}_{\boldsymbol{z}_{/i}}[g^{(2)}(\boldsymbol{z}_{/i}) - g^{(1)}(\boldsymbol{z}_{/i})] = \text{VAR}_{\boldsymbol{z}_{/i}}[g^{(2)}(\boldsymbol{z}_{/i})] + \text{VAR}_{\boldsymbol{z}_{/i}}[g^{(1)}(\boldsymbol{z}_{/i})] - 2\text{COV}_{\boldsymbol{z}_{/i}}[g^{(1)}(\boldsymbol{z}_{/i}), g^{(2)}(\boldsymbol{z}_{/i})],$$

$$\text{VAR}_{\boldsymbol{z}_{/i}, \boldsymbol{z}'_{/i}}[g^{(2)}(\boldsymbol{z}_{/i}) - g^{(1)}(\boldsymbol{z}'_{/i})] = \text{VAR}_{\boldsymbol{z}_{/i}}[g^{(2)}(\boldsymbol{z}_{/i})] + \text{VAR}_{\boldsymbol{z}_{/i}}[g^{(1)}(\boldsymbol{z}_{/i})]. \tag{27}$$

If the functions $g^{(a)}(\boldsymbol{z}_{/i})$ are highly correlated the ARM estimator will have a much higher variance than RAM.

for multivariate functions compared to RAM. The ARM estimator has straightforward extensions to hierarchical $q(z)$ (similar to Eq. (4)) and to categorical variables Yin & Zhou (2018).

## C  Improved CR estimator for Bayesian networks

In this appendix we derive the improved CR estimator for a hierarchical $q_\phi(z) = \prod_i q_{\phi,i}(z_i|z_{<i})$. For simplicity, we do this for two variables $q_\phi(z_1, z_2) = q_{\phi,2}(z_2|z_1)q_{\phi,1}(z_1)$ with the gradient given in Eq. (4):

$$\partial_\phi \mathcal{L} = \partial_\phi q_{\phi,1} \left[ \sum_{z_2} q_{\phi,2}(z_2|1)f(1, z_2) - \sum_{z_2} q_{\phi,2}(z_2|0)f(0, z_2) \right]$$
$$+ \sum_{z_1} q_{\phi,1}(z_1)\partial_\phi q_{\phi,2}(1|z_1) \left[ f(z_1, 1) - f(z_1, 0) \right]$$

where $q_{\phi,1} = q_{\phi,1}(z_1 = 1)$. We denote the two contributions to $\partial_\phi \mathcal{L}$ by $\partial_\phi \mathcal{L}^{(1,2)}$ and determine them separately. Using the integral trick (10) the second contribution can be written as

$$\partial_\phi \mathcal{L}^{(2)} = \sum_{z_1} q_{\phi,1}(z_1)\partial_\phi q_{\phi,2}(1|z_1) \int d\rho_1 \frac{\partial \zeta_2}{\partial \rho_2} \partial_{\zeta_2} f(z_1, \zeta_2) \approx \mathbb{E}_\rho \left[ \partial_\phi q_{\phi,2}(1|\zeta_1) \frac{\partial \zeta_2}{\partial \rho_2} \partial_{\zeta_2} f(\zeta_1, \zeta_2) \right] \tag{28}$$

where $\zeta_2 = g(\rho_2, q_2(1|\zeta_1))$ and $\zeta_1 = g(\rho_1, q_1)$. The first term can be handled similarly:

$$\partial_\phi \mathcal{L}^{(1)} = \partial_\phi q_{\phi,1} \int d\rho_1 \frac{\partial \zeta_1}{\partial \rho_1} \partial_{\zeta_1} \left( \sum_{z_2} q_{\phi,2}(z_2|\zeta_1)f(\zeta_1, z_2) \right) \approx \mathbb{E}_\rho \left[ \partial_\phi q_{\phi,1} \frac{\partial \zeta_1}{\partial \rho_1} \partial_{\zeta_1} f(\zeta_1, \zeta_2) \right] \tag{29}$$

Combining these contributions we arrive at an expression very similar to the reparameterization trick with the replacement $\partial_{q_i} \zeta_i \to \partial_{\rho_i} \zeta_i$:

$$\partial_\phi \mathcal{L} \approx \partial_\phi \mathbb{E}_\rho \left[ f(\zeta_1, \zeta_2) \right]_{\partial_{q_i}\zeta_i \to \partial_{\rho_i}\zeta_i} \tag{30}$$

## D  Categorical PWL estimator

Here, we derive the PWL estimator for categorical variables. The derivative for a single categorical variable $y = (y^0, ... y^{A-1})$ with $y^a \in \{0, 1\}$ and $\sum_a y^a = 1$ is given in Eq. (5) as

$$\partial_\phi \mathcal{L} = \partial_\phi \sum_y q_\phi(y)f(y) = \sum_a \partial_\phi l_\phi^a \sum_b q^a q^b (f^a - f^b), \tag{31}$$

We again apply the integral trick (10) to represent the difference $f^a - f^b$. To do that we relax the variable $y$ so that it interpolates between $(y^a, y^b) = (1, 0)$ and $(y^a, y^b) = (0, 1)$ as $y \to y^{a,b} = \left\{ y^a = \left[ 0.5 + \alpha^{a,b}(\rho^{a,b} - q^b/(q^a + q^b)) \right]_0^1, y^b = 1 - y^a, y^{c \neq a,b} = 0 \right\}$ where $\rho^{a,b} \in \mathcal{U}[0, 1]$ and $\alpha^{a,b}$ is the slope. The relaxed objective then takes the form

$$\mathcal{L} = \int_0^1 \prod_{a<b} d\rho^{a,b} \sum_{a<b} w^{a,b} f(y^{a,b}), \tag{32}$$

where $w^{a,b}$ are weights to be determined. The gradient of this relaxed objective with respect to the logit $l^a$ is

$$\partial_{l^a} \mathcal{L} = \mathbb{E}_\rho \left[ \sum_{b \neq a} w^{a,b} \partial_{\rho^{a,b}} f(y^{a,b}) \frac{q^a q^b}{(q^a + q^b)^2} \right] = \sum_{b \neq a} w^{a,b} \frac{q^a q^b}{(q^a + q^b)^2} (f^a - f^b). \tag{33}$$

Comparison with Eq. (31) gives $w^{a,b} = (q^a + q^b)^2$. However, computing the sum over the edges of the simplex is prohibitively expensive, so we choose to replace it with sampling from the set of edges with probability $p^{a,b} = (q_a + q_b)/(\sum_{a<b} q^a + q^b) = (q^a + q^b)/(A - 1)$. The reason for choosing this distribution is that Eq. (32) must give correct value of the objective (not just the derivative) as

the relaxation parameter $\beta \to \infty$, which requires the probability of each state $y^a = 1$ to be equal to $q^a$. For relaxed edge $(a, b)$ the probability of $y^a = 1$ is equal to $q^a/(q^a + q^b)$, and thus the total probability of $y^a = 1$ is:

$$\sum_{b \neq a} p^{a,b} \frac{q^a}{q^a + q^b} = \sum_{b \neq a} \frac{q^a + q^b}{A - 1} \frac{q^a}{q^a + q^b} = q^a. \tag{34}$$

Finally, to get the correct weights $w^{a,b}$ we must rescale each term by a factor $\gamma^{a,b} = (A-1)(q^a+q^b)$ so that $w^{a,b} = \gamma^{a,b} p^{a,b}$. In summary, the relaxed objective has the following form:

$$\mathcal{L} = \mathbb{E}_{(a,b) \sim p^{a,b}} \left[ \mathbb{E}_{\rho \in \mathcal{U}[0,1]} \left[ f(\tilde{y}^{a,b}) \right] \right], \tag{35}$$

where $\tilde{y}^{a,b}$ has the same value as $y^{a,b}$ but has the gradient scaled by $\gamma^{a,b} = (A-1)(q^a + q^b)$.

## E   REBAR

In this section, we derive a simple expression for REBAR gradients Tucker et al. (2017) and describe its connection to ICR estimators. We also present additional experimental results for VAE training with REBAR for different continuous relaxations.

REBAR gradients of the objective (1) can be written as:

$$\partial_\phi \mathcal{L} = \mathbb{E}_{q_\phi(\boldsymbol{z}) q_\phi(\boldsymbol{\zeta}|\boldsymbol{z})} \left[ \partial_\phi \log q_\phi(\boldsymbol{z}) \left( f(\boldsymbol{z}) - f(\boldsymbol{\zeta}) \right) \right] + \partial_\phi \mathbb{E}_{q_\phi(\boldsymbol{\zeta})} \left[ f(\boldsymbol{\zeta}) \right] - \partial_\phi \mathbb{E}_{q_{\mathbf{sg}(\phi)}(\boldsymbol{z}) q_\phi(\boldsymbol{\zeta}|\boldsymbol{z})} \left[ f(\boldsymbol{\zeta}) \right],$$

where $\mathbf{sg}(\phi)$ denotes "stop gradient", and distributions $q_\phi(\boldsymbol{\zeta})$ and $q_\phi(\boldsymbol{\zeta}|\boldsymbol{z})$ are reparameterizable. Using explicit reparameterizations for $\boldsymbol{\zeta}(\boldsymbol{\rho}, q_\phi)$ and $\boldsymbol{\zeta}(\mathbf{u}, q_\phi|\boldsymbol{z})$ with $\mathbf{u} \in \mathcal{U}[0, 1]$, we can rewrite the gradient as

$$\partial_\phi \mathcal{L} = \mathbb{E}_{q_\phi(\boldsymbol{z}) q_\phi(\boldsymbol{\zeta}|\boldsymbol{z})} \left[ \partial_\phi \log q_\phi(\boldsymbol{z}) \left( f(\boldsymbol{z}) - f(\boldsymbol{\zeta}) \right) \right] + \mathbb{E}_{\boldsymbol{\rho}} \left[ \partial_{\boldsymbol{\zeta}} f(\boldsymbol{\zeta}) \partial_\phi \boldsymbol{\zeta}(\boldsymbol{\rho}, q_\phi) \right] - \mathbb{E}_{q_\phi(\boldsymbol{z}),\mathbf{u}} \left[ \partial_{\boldsymbol{\zeta}} f(\boldsymbol{\zeta}) \partial_\phi \boldsymbol{\zeta}(\mathbf{u}, q_\phi|\boldsymbol{z}) \right]. \tag{36}$$

Focusing on binary variables, the reparameterization for the conditional distribution has the form $\boldsymbol{\zeta}(\mathbf{u}, q_\phi|\boldsymbol{z}) = \boldsymbol{\zeta}(\tilde{\boldsymbol{\rho}}, q_\phi)$, where $\tilde{\boldsymbol{\rho}} = 1 - q_\phi + \mathbf{u} \left( \boldsymbol{z} q_\phi - (1 - \boldsymbol{z})(1 - q_\phi) \right)$. In order to minimize the variance, it is preferable to tie the parameters $\boldsymbol{\rho}$ and $(\boldsymbol{z}, \mathbf{u})$. This can be achieved by setting $\boldsymbol{z} = \Theta(\boldsymbol{\rho} - 1 + q_\phi)$, and choosing

$$u = \begin{cases} \frac{\boldsymbol{\rho} - 1 + q_\phi}{q_\phi} & \text{if } z = 1, \\ -\frac{\boldsymbol{\rho} - 1 + q_\phi}{1 - q_\phi} & \text{if } z = 0. \end{cases} \tag{37}$$

This is equivalent to sampling $(\boldsymbol{z}, \boldsymbol{\zeta}) \sim q_\phi(\boldsymbol{z}) q_\phi(\boldsymbol{\zeta}|\boldsymbol{z})$ as $\boldsymbol{z} = \Theta(\boldsymbol{\rho} - 1 + q_\phi), \boldsymbol{\zeta} = \boldsymbol{\zeta}(\tilde{\boldsymbol{\rho}}, q_\phi)$ where

$$\tilde{\boldsymbol{\rho}} = 1 - q_\phi + \mathbf{sg}(\boldsymbol{\rho} - 1 + q_\phi) \left( \boldsymbol{z} \frac{q_\phi}{\mathbf{sg}(q_\phi)} + (1 - \boldsymbol{z}) \frac{1 - q_\phi}{1 - \mathbf{sg}(q_\phi)} \right).$$

This way $\tilde{\boldsymbol{\rho}} = \boldsymbol{\rho}$ by value but $\tilde{\boldsymbol{\rho}}$ does depend on $\phi$ such that $\boldsymbol{\zeta}(\tilde{\boldsymbol{\rho}}, q_\phi)$ has the correct gradient w.r.t. $\phi$ that follows from the conditional distribution $q_\phi(\boldsymbol{\zeta}|\boldsymbol{z})$. We can rewrite (36) as

$$\partial_\phi \mathcal{L} = \mathbb{E}_{q_\phi(\boldsymbol{z}) q_\phi(\boldsymbol{\zeta}|\boldsymbol{z})} \left[ \partial_\phi \log q_\phi(\boldsymbol{z}) \left( f(\boldsymbol{z}) - f(\boldsymbol{\zeta}) \right) \right] + \mathbb{E}_{\boldsymbol{\rho}} \left[ \partial_{\boldsymbol{\zeta}} f(\boldsymbol{\zeta}) \left( \partial_\phi \boldsymbol{\zeta}(\boldsymbol{\rho}, q_\phi) - \partial_\phi \boldsymbol{\zeta}(\tilde{\boldsymbol{\rho}}, q_\phi) \right) \right]. \tag{38}$$

Since the explicit dependence of $\boldsymbol{\zeta}(\boldsymbol{\rho}, q_\phi)$ and $\boldsymbol{\zeta}(\tilde{\boldsymbol{\rho}}, q_\phi)$ on $q_\phi$ is the same, we have $\partial_\phi \boldsymbol{\zeta}(\boldsymbol{\rho}, q_\phi) - \partial_\phi \boldsymbol{\zeta}(\tilde{\boldsymbol{\rho}}, q_\phi) = -\partial_\phi \boldsymbol{\zeta}(\tilde{\boldsymbol{\rho}}, \mathbf{sg}(q_\phi))$ and we arrive at our final expression for the REBAR gradient:

$$\partial_\phi \mathcal{L} = \mathbb{E}_{\boldsymbol{\rho}, \boldsymbol{z} = \Theta(\boldsymbol{\rho} - 1 + q_\phi)} \left[ \partial_\phi \log q_\phi(\boldsymbol{z}) \left( f(\boldsymbol{z}) - f(\boldsymbol{\zeta}(\boldsymbol{\rho}, q_\phi)) \right) - \partial_\phi f(\boldsymbol{\zeta}(\tilde{\boldsymbol{\rho}}, \mathbf{sg}(q_\phi))) \right]. \tag{39}$$

The advantage of Eq. (39) is two-fold: first, it allows for a simple implementation valid for any continuous relaxation and uses only two function evaluations. Second, it gives a suggestive relation to the ICR estimator: indeed using $\partial_\phi \tilde{\boldsymbol{\rho}} = \partial_\phi q_\phi \left( -1 + \boldsymbol{z} \frac{\boldsymbol{\rho} - 1 + q_\phi}{q_\phi} - (1 - \boldsymbol{z}) \frac{\boldsymbol{\rho} - 1 + q_\phi}{1 - q_\phi} \right)$ we can write REBAR gradient as

$$\partial_\phi \mathcal{L}_{\text{REBAR}} = \partial_\phi \mathcal{L}_{\text{ICR}} + R_1 + R_2$$
$$\partial_\phi \mathcal{L}_{\text{ICR}} = \mathbb{E}_{\boldsymbol{\rho}} \left[ \partial_\phi q_\phi \partial_{\boldsymbol{\rho}} \boldsymbol{\zeta} \partial_{\boldsymbol{\zeta}} f(\boldsymbol{\zeta}) \right]$$
$$R_1 = \mathbb{E}_{\boldsymbol{\rho}, \boldsymbol{z} = \Theta(\boldsymbol{\rho} - 1 + q_\phi)} \left[ -\partial_\phi q_\phi \partial_{\boldsymbol{\rho}} \boldsymbol{\zeta} \partial_{\boldsymbol{\zeta}} f(\boldsymbol{\zeta}) \left( \boldsymbol{z} \frac{\boldsymbol{\rho} - 1 + q_\phi}{q_\phi} - (1 - \boldsymbol{z}) \frac{\boldsymbol{\rho} - 1 + q_\phi}{1 - q_\phi} \right) \right]$$
$$R_2 = \mathbb{E}_{\boldsymbol{\rho}, \boldsymbol{z} = \Theta(\boldsymbol{\rho} - 1 + q_\phi)} \left[ \partial_\phi \log q_\phi(\boldsymbol{z}) \left( f(\boldsymbol{z}) - f(\boldsymbol{\zeta}(\boldsymbol{\rho}, q_\phi)) \right) \right]. \tag{40}$$

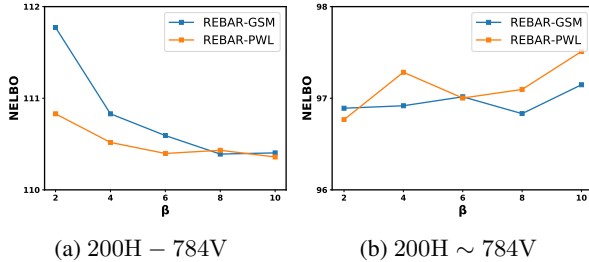

(a) $200H - 784V$         (b) $200H \sim 784V$

Figure 6: MNIST training with REBAR using GSM and PWL relaxations, dependence of the final trained NELBO on $\beta$: (a) the linear architecture. (b) the non-linear architecture.

Here $R_{1,2}$ are correction terms. The term $R_2$ can have high variance due to the fact that $\mathrm{var}(\partial_{q_\phi} \log q_\phi(z)) \sim \frac{1}{q_\phi(1-q_\phi)}$ while $R_1$ has variance comparable to $\partial_\phi \mathcal{L}_{\mathrm{ICR}}$ since $\mathrm{var}(z\frac{\rho-1+q_\phi}{q_\phi} - (1-z)\frac{\rho-1+q_\phi}{1-q_\phi}) \sim O(1)$. In summary REBAR estimator can be viewed as ICR plus corrections that remove the bias of ICR estimator. In practice the correction term $R_2$ can increase the variance of REBAR estimator compared to ICR and slow down the training.

We now perform the comparison of two types of continuous relaxations used by REBAR estimator. We train VAE as in Section 4.2 with $200H - 784V$ and $200H \sim 784V$ architectures using REBAR with GSM and PWL relaxations. Fig. 6 shows the dependence of the final NELBO on $\beta$. In the case of the linear mode $200H - 784V$ we find that PWL relaxation is slightly more stable while for the non-linear model the two relaxations perform similarly.

## F  EXPERIMENTAL DETAILS

For the toy example of Section 4.1 we optimize $\mathcal{L}$ with respect to $q_\phi(z = 1)$ using Adam (Kingma & Ba (2014)) for 2000 iterations with learning rate $r = 0.01$ using minibatches of size 100 to reduce the variance of gradients. We initialize $q_\phi(z = 1)$ to the wrong maximum of the relaxed function by setting $q_\phi(z = 1) = \sigma(5)$.

For VAE training experiments, following Maddison et al. (2016); Jang et al. (2016); Tucker et al. (2017), we consider four architectures with binary variables denoted by $200H - 784V$, $200H - 200H - 784V$, $200H \sim 784V$, and $200H \sim 200H \sim 784V$. Here $-$ denotes a linear layer, while $\sim$ denotes two layers of 200 hidden units with $\tanh$ nonlinearity and batch normalization. In our experiments we use the Adam optimizer with default parameters and a fixed learning rate 0.0003, run for $4 \cdot 10^6$ steps with minibatches of size 100. We repeat each experiment 5 times with random seed and plot the mean.

## G  VAE ENCODER TRAINING

During training of VAEs, the encoder and decoder models interact in complex ways. To eliminate these effects to more directly identify the impact of better gradients, we use a fixed pre-trained decoder, and minimize Eq. (21) with respect to encoder parameters $\phi$ only. As shown in Fig. 7, the conclusions from joint training remain unchanged: GSM underperforms due to its bias. Interestingly however, RAM still overfits early in training for the $200H \sim 784V$ model, showing that better gradients can negatively affect optimization.

## H  OPTIMIZATION

We apply gradient estimators for discrete optimization. Following Patish & Ullman (2018) we study the NP-hard task of finding a maximal clique in a graph. To model this problem, the binary variable $z_i = 1/0$ indicates the presence/absence of vertex $i$ in a maximal clique. If $A_{i,j}$ is the adjacency matrix of the graph and $d = \sum_i z_i$ is the size of the clique, then the objective function considered

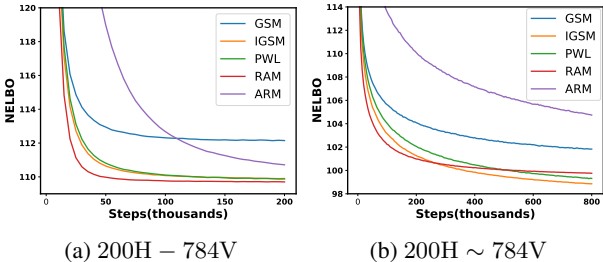

(a) $200H - 784V$    (b) $200H \sim 784V$

Figure 7: Encoder training the with fixed pre-trained decoder on MNIST: (a) the linear architecture. (b) the non-linear architecture.

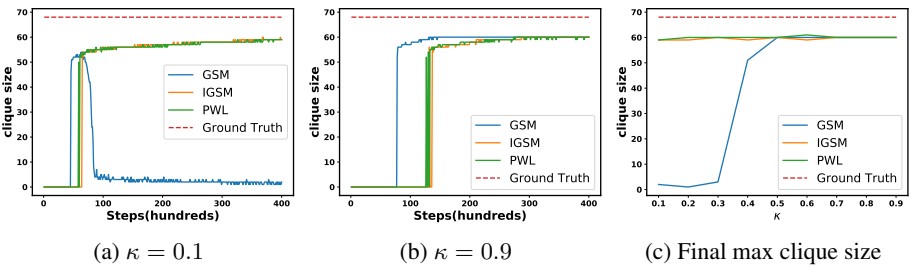

(a) $\kappa = 0.1$    (b) $\kappa = 0.9$    (c) Final max clique size

Figure 8: Finding maximal cliques in the $C1000.9$ graph: (a) and (b) show the course of optimization for two settings of $\kappa$ and different gradient estimators. (c) plots the final maximal clique size across $\kappa$ for the estimators.

in Patish & Ullman (2018) is

$$f(z) = -\frac{\sum_{ij} z_i A_{ij} z_j}{d(d - 1 + \kappa)}, \tag{41}$$

where $\kappa \in [0,1]$ is a hyperparameter. Patish & Ullman (2018) use factorial $q_\phi(z)$ to minimize $\mathcal{L}[\phi] = \sum_z q_\phi(z)f(z)$. At the end of optimization $q_\phi(z)$ typically collapses to a single state corresponding to a clique. Most often this clique is a local minimum and not the maximal clique.

We illustrate the performance of CR estimators on a particular graph (1000 nodes, 450000 edges), labeled $C1000.9$ from the DIMACS data set Johnson & Trick (1996). We minimize $f(z)$ using Adam with default settings, learning rate 0.01 for 40000 iterations. We perform 1000 minimizations in parallel and choose the best clique found at each iteration. Fig. 8(a),(b) show the size of clique found by each of the estimators for two values of the hyperparameter $\kappa = 0.1, 0.9$. At $\kappa = 0.1$ the bias of GSM causes trapping in the wrong minima but ICR converges to a good local minimum. In contrast, at $\kappa = 0.9$ the GSM bias accelerates convergence to a good local minimum. Fig. 8(c) shows the dependence of the final clique size on $\kappa$; unbiased estimators provide more stable results.

# I    ADDITIONAL EXPERIMENTS

## I.1    BINARY CONCAVE TOY EXAMPLE

We evaluate performance on the concave function $f(\zeta) = -(\zeta - 0.45)^2$ shown in Fig. 9(a). The training setup is identical to Section 4.1 but with $q_\phi(z = 1)$ initialized to $\sigma(-5)$. Fig. 9(b) demonstrates that the GSM optimization gets trapped in the wrong minimum due to its bias towards the dominant mode $\zeta = 0$.

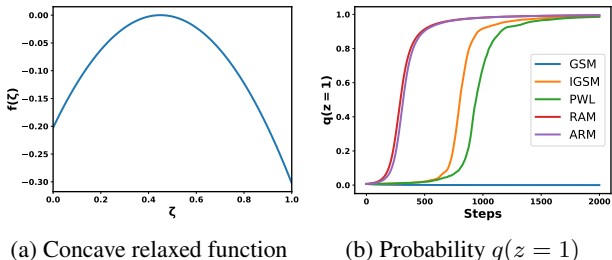

(a) Concave relaxed function      (b) Probability $q(z = 1)$

Figure 9: Single binary variable concave toy example. (a) Relaxed function $f(\zeta) = -(\zeta - 0.45)^2$ (b) Probability $q_\phi(z = 1)$ during optimization

## I.2 CATEGORICAL TOY EXAMPLE

We consider a categorical example with convex and concave functions of a single categorical variable $y$ having 10 values. We take $f(y) = \pm \sum_a (g^a - y^a)^2$ such that $g^0 = 0.9, g^1 = 1.1$, and $g^{i>1} = 1$. The convex function has a minimum at $y^1 = 1$, while the concave function is minimized at $y^0 = 1$. We compare 4 estimators for minimizing this function: RAM of Eq. (5), PWL of Eq. (18), GSM of Eq. (19) and IGSM of Eq. (20). The probability of the true minimum is shown in Fig. 10. In both the convex and concave cases the GSM estimator exhibits a bias preventing it from finding the minimum. IGSM is less biased than GSM which allows it to find the true minimum. The PWL estimator is unbiased but has higher variance then IGSM which slows down its optimization.

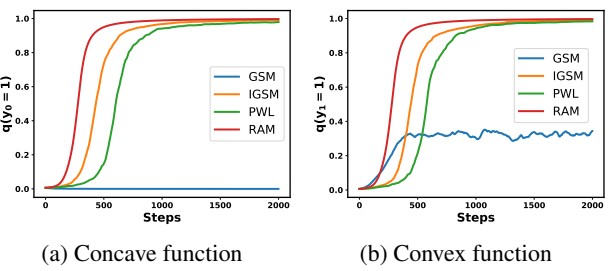

(a) Concave function      (b) Convex function

Figure 10: The probability of the true minimum for a single categorical-variable toy example with (a) concave function $f(y) = -\sum_a (g^a - y^a)^2$ and (b) convex function $f(y) = \sum_a (g^a - y^a)^2$

## I.3 DISCRETE VARIATIONAL AUTOENCODERS

Fig. 11 shows the results for all 4 architectures $200H - 784V$, $200H - 200H - 784V$, $200H \sim 784V$, and $200H \sim 200H \sim 784V$ at $\beta = 2$. Hierarchical models with two layers of latent units in Fig. 11(b),(d) exhibit similar trends to the factorial case considered in the main text. The GSM estimator converges to a higher NELBO in the linear case and becomes unstable in the non-linear case.

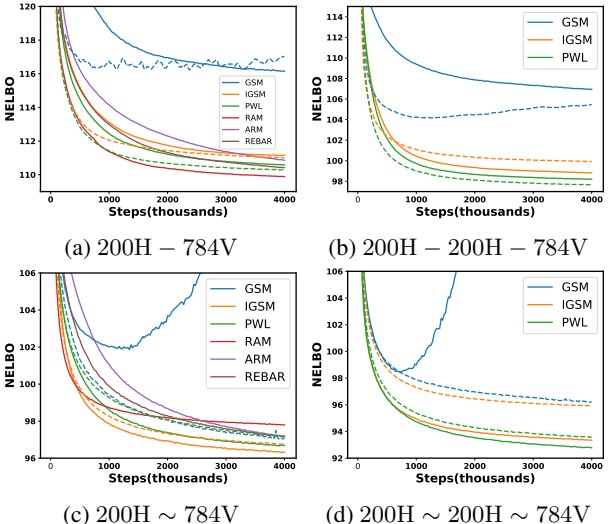

(a) $200H - 784V$  (b) $200H - 200H - 784V$

(c) $200H \sim 784V$  (d) $200H \sim 200H \sim 784V$

Figure 11: MNIST training (CR estimators use $\beta = 2$). Solid/dashed lines correspond to one/two-pass training.

We also compare the performance of GSM and IGSM estimators in the categorical case $20 \times 10H - 784V$ with latent space being 20 categorical variables with 10 classes Fig. 12. We see that IGSM outperforms GSM estimator when using two-pass training. This indirectly confirms that IGSM is less biased. The PWL estimator performs best indicating the advantages ICR estimators.

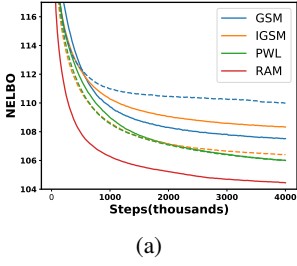

(a)

Figure 12: MNIST training on categorical linear architecture $20 \times 10H - 784V$ (CR estimators use $\beta = 2$). Solid/dashed lines correspond to one/two-pass training.

For completeness we also show the training curves for OMNIGLOT dataset at $\beta = 2$ in Fig 13. Interestingly, one-pass training performs better in all cases. PWL estimator performs best among considered CR estimators.

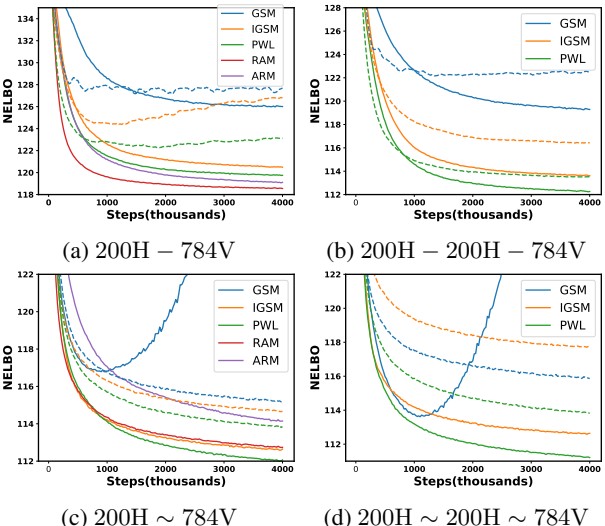

(a) $200H - 784V$

(b) $200H - 200H - 784V$

(c) $200H \sim 784V$

(d) $200H \sim 200H \sim 784V$

Figure 13: OMNIGLOT training (CR estimators use $\beta = 2$). Solid/dashed lines correspond to one/two-pass training.

