# OpenReview forum: "Improved Gradient Estimators for Stochastic Discrete Variables"
_ICLR.cc/2019/Conference_

### Official Review · AnonReviewer1 · 2018-11-01
**A paper reviewing and improving different types of gradient estimators**

**Rating:** 6
**Confidence:** 4

**Review:**

The papers studies estimators of gradients taken from expectations with respect to the distribution parameters. The paper has studied two main types of estimators, Finite Difference and Continuous Relaxation. The paper made several improvements to existing estimators.

My rating of the paper in different aspects (quality 6, clarity 8, originality 6, significance 4).

Pros:
1. The paper has made a nice introduction of FD and CR estimators. The improvements over previous estimators are concrete -- it is generally clear to see the benefit of these improvements.

2. The first method reduces the running time of the RAM estimator. The second method (IGM) reduces the bias of GM estimator. The first improvement avoids many function evaluations when the probability is extreme. The second improvement helps to correct bias introduced by continuous approximation of \zeta_i itself.

Cons:
1. the paper content is a little disjointed: the improvement over RAM has not much relation with later improvements. It seems the paper is stacking different things into the paper.

2. All these improvements are not very significant considering a few previous papers on this topic. Some arguments are not rigorous. (see details below)

3. A few important papers are not well discussed and omitted from the experiment section.

Detailed comments

1. The REBAR estimator [Tucker et al., 2017] and the LAX estimator [Grathwohl et al., 2018] use continuous approximation and correct it to be unbiased. These papers in this thread are not well discussed in the paper. They are not compared in the experiment either.

2. In the equation 7 and above: what does 4 mean? When beta \neq 4, do you still get unbiased estimation? My understanding is that the estimator is unbiased only when beta=4. (correct me if I'm wrong)

3. The paper argues that the variance of the estimator is mostly decided by the variance of q(zeta)^-1 when the function is smooth. I feel this argument is not very clear. First, what do you mean by saying the function is smooth? The derivative is near a constant in [0, 1]?

4. In the PWL development, the paper argues that we can choose alpha_i \approx 1/(q_i(1-q_i)) to minimize the variance. However, my understanding is, the smaller alpha_i, the smaller variance.

---

> ### Author Response · Authors · 2018-11-15
> **Response to Reviewer 1**
>
> Thank you for the helpful suggestions. We have updated the paper. Please find our responses below:
>
> "1. the paper content is a little disjointed: the improvement over RAM has not much relation with later improvements. It seems the paper is stacking different things into the paper." - We agree that improvement to RAM has little to do with CR estimators discussed afterwards. However, sampled RAM does have a connection to CR: both estimators evaluate the gradient only through a subset of stochastic variables $z_i$. Sampled RAM chooses this subset explicitly and evaluates the gradients via FD.  CR samples relaxed variables and effectively only a subset of them will deviate from {0,1} (in the binary variable case) and will possess non-zero gradients. We have added a clarifying sentence to the text.
>
> "1. The REBAR estimator [Tucker et al., 2017] and the LAX estimator [Grathwohl et al., 2018] use continuous approximation and correct it to be unbiased. These papers in this thread are not well discussed in the paper. They are not compared in the experiment either." - We have added a comparison to REBAR in Fig. 4, 5 and Appendix E.
>
>
> "2. In the equation 7 and above: what does 4 mean? When beta \neq 4, do you still get unbiased estimation? My understanding is that the estimator is unbiased only when beta=4. (correct me if I'm wrong)" - We added a factor of 4 so that the probability of keeping a variable, $p = [4 q (1-q)]/\beta$,  is consistent with CR estimators. For example, in the case of the PWL estimator we have chosen to parameterize the slope as $\alpha = \beta/[4 q(1-q)]$ (above Eq (16)) which leads to the probability for the variable to have non-zero gradient equal to $p = [4 q (1-q)]/\beta$.  Other than that there is no special meaning to having it there. Eq (7) contains the factor $\beta/4$ just to compensate for our parameterization of $p$, so that $\E[ \beta \zeta/4] = q (1-q)$ and on average Eq. (7) gives the same gradient as Eq. (3). So Eq. (7) is unbiased for every value of $\beta$.
>
>
> "3. The paper argues that the variance of the estimator is mostly decided by the variance of q(zeta)^-1 when the function is smooth. I feel this argument is not very clear. First, what do you mean by saying the function is smooth? The derivative is near a constant in [0, 1]?" - We agree that this statement about smoothness of function $f(\zeta)$ is a bit vague. We have replaced it in the paper with “If the derivative $\partial_\zeta f(\zeta)$ does not change significantly in the interval $\zeta \in [0, 1]$ then the variance of  this  estimate  is  controlled  by ... ”.
>
> "4. In the PWL development, the paper argues that we can choose alpha_i \approx 1/(q_i(1-q_i)) to minimize the variance. However, my understanding is, the smaller alpha_i, the smaller variance." - Thank you for pointing this out! We agree that in order to minimize the variance one has to minimize variance of each term independently, which can be done by choosing the smallest possible $\alpha_i$. We have removed this argument from the paper.

---

### Official Review · AnonReviewer3 · 2018-11-02
**The paper proposed to reduce the computation of the Re-parameterization and Marginalization method and the bias of Continuous Relaxation estimator.**

**Rating:** 6
**Confidence:** 3

**Review:**

The paper proposed a modification to RAM that allows us to trade decreased computational cost for increased variance. It also proposes an improved continuous relaxation (ICR) estimator to reduce the bias of CR, which is extended to categorical variables.
The proposed piece-wise linear relaxation (PWL) can be considered as the inverse CDF of the random variable is very interesting. The ICR estimators can also be extended to categorical variables.
The paper is well written. I have some questions:
1.	How does the dimension of the variables affect the bias and variance of the proposed estimator?
2.	Dose the proposed estimators applicable to hierarchical models with multi-discrete latent variables?
3.	 What’s the performance of the proposed method compared with the others in terms of running time?

---

> ### Author Response · Authors · 2018-11-15
> **Response to Reviewer 3**
>
> Thank you for the clarifying questions. Please find our responses below:
>
> 1. "How does the dimension of the variables affect the bias and variance of the proposed estimator?" - For the case of a factorial posterior distribution over binary variables, our proposed improved estimator is given by Eq. (13). The bias here comes from the the deviation of relaxed $\zeta_{\setminus i}$ from binary $z_{\setminus i}$. The magnitude of this bias depends on the function $f(z)$ that we are minimizing, but in general the bias is expected to grow with the number of variables. The main contribution to the variance of Eq. (13) comes from the terms $\partial  \zeta_i / \partial \rho_i$. Since these terms are independent (they depend on different $\rho_i \in U[0,1]$), the variance of the sum is expected to grow linearly with $M$ in general. This means that relative standard deviation (standard deviation / mean) scales as $\sim 1/\sqrt{M}$ in general.
>
> 2. "[Are] the proposed estimators applicable to hierarchical models with multi-discrete latent variables?"  - We show in the Appendix C, the proposed improved estimators can be applied to hierarchical models (Bayesian network distributions): one just needs to replace $\partial_{q_i} \zeta_i$ with $\partial_{\rho_i} \zeta_i$ throughout the hierarchy.
>
> 3. "What's the performance of the proposed method compared with the others in terms of running time?" - We do not report running times in the paper, but in our experiments we saw similar running times for improved continuous relaxations versus the original ones (single function evaluation). In the case of finite-difference estimators, RAM involves $M$ function evaluations, Sampled RAM uses varying number of function evaluations (roughly $ M/10$ by the end of training), ARM uses 2 function evaluations. However, due to the GPU parallelization and small size of the neural networks in this work, we do not observe a significant variation in running time.

---

### Official Review · AnonReviewer2 · 2018-11-03
**Well written; contributions intuitively explained and motivated**

**Rating:** 7
**Confidence:** 4

**Review:**

After revision:
The authors have addressed all points in my review. Although I will not be increasing the score, these fixes certainly increase the confidence of my evaluation and I think it deserves to be accepted.

====================

Summary: This paper analyzes finite-difference and continuous relaxation gradient estimators for discrete random variables and from their analysis develop improvements to these existing methods. They empirically demonstrate the improvement by evaluating the gradient estimators on toy tasks and an autoencoding task.

Writing: I found this paper very well written and explained. It covered an extensive background concisely while introducing all necessary ideas to understand the contributions of the paper.

Comments: Overall, I found the ideas presented in this paper interesting and novel, and results sufficiently strong to support the ideas. Though the contributions are not groundbreaking, they will certainly be useful to researchers in this space. I have some minor comments relating to notation and related work.

- I found the notation in Section 3.2 to be a little confusing, namely that $\zeta$ appears as both a random variable and a continuous function (that takes in one variable in the paragraph after eq11, but takes in two variables in eq15). I understand that the authors may have done this to suppress extra notation, but I found this section harder to understand than the rest due to this choice. There is also a small typo in eq2 where the $\phi$ from $l_\phi$ is dropped.

- I think it would be useful to compare IGSM and PWL against a score-function gradient estimator (maybe REBAR, given the similarity in experiment setup). The authors do contextualize the line of work concerning score-function gradient estimators. However, since SF estimators are unbiased but high variance and the authors aim to reduce bias at the cost of variance, I think evaluating SF baselines will better contextualize the tradeoffs made in this paper.

---

> ### Author Response · Authors · 2018-11-15
> **Response to Reviewer 2**
>
> Thank you for the constructive comments and suggestions. We have updated the paper:
> - We clarified the notation in Section 3.2, by removing this ambiguity of using $\zeta$ symbol.
> - We also corrected the typo $l$ -> $l_\phi$ (thank you for pointing it out!).
> - We added experimental results using REBAR in training VAE in Fig. 4, 5 and Appendix E. We agree that these experiments   provide another useful  comparison with the state-of-the-art.

---

### Meta-Review · Area_Chair1 · 2018-12-14
**Sensible ideas scattered throughout, but does not engage with similar earlier work.**

**Recommendation:** Reject
**Confidence:** 3

**Metareview:**

Strengths: This paper provides a useful review of some of the recent work on gradient estimators for discrete variables, and proposes both a computationally more efficient variant of one, and a new estimator based on piecewise linear functions.

Weaknesses:  Many new ideas are scattered throughout the paper.  The notation is a bit dense.  Comparisons to RELAX, which had better results than REBAR, are missing.  Finally, it seems that REBAR was trained with a fixed temperature, instead of optimizing it during training, which is one of the main benefits of the method.

Points of contention: Only R1 mentioned the omission of REBAR and RELAX.  A discussion and a few comparisons to REBAR were added to the paper, but only in a few experiments.

Consensus:  This paper is borderline.  I agree with R1: quality 6, clarity 8, originality 6, significance 4.  All reviewers agreed that this was a decent paper but I think that R2 and R3 were relatively unfamiliar with the existing literature.

Update for clarification:
=====================

This section has been added to clarify the reasons for rejection.  The abstract of the paper states:

"We show that the commonly used Gumbel-Softmax estimator is biased and propose a simple method to reduce it. We also derive a simpler piece-wise linear continuous relaxation that also possesses reduced bias. We demonstrate empirically that reduced bias leads to a better
performance in variational inference and on binary optimization tasks."

The fact that Gumbel-Softmax is biased is well-known.  Reducing its bias was the motivation for developing the _exactly_ unbiased REBAR method, which already has similar asymptotic complexity.  A major side-benefit of using an exactly unbiased estimator is that the estimator's hyperparameters can be automatically tuned to reduce variance, as in REBAR and RELAX.

This paper focuses on methods for reducing bias and variance, but hardly discusses related methods that already achieved its stated aims. This a major weakness of the paper.  The experiments only compared with REBAR, and did not even tune the temperature to reduce variance (removing one of its major advantages).

This reject decision is not made mainly on lack of experiments or state-of-the-art results.  It's because the idea of reducing the bias of continuous-relaxation-based gradient estimators has already been fruitfully explored, and zero-bias CR estimators have been developed, but this work mostly ignores them.  However, thorough experiments are always going to be necessary for a paper proposing biased estimators, because there are already many such estimators, and little theory to say which ones will work well in which situations.

Suggestions to improve the paper:  Run experiments on all methods that directly measure bias and variance.  Incorporate discussion of REBAR throughout, not just in an appendix.  Run comparisons against REBAR and RELAX without crippling their ability to reduce variance.   Do more to characterize when different estimators will be expected to be effective.